# Protein Condensate Atlas from predictive models of heteromolecular condensate composition

Kadi L. Saar [1,2] ✉, Rob M. Scrutton [2,3], Kotryna Bloznelyte[1], Alexey S. Morgunov[2], Lydia L. Good [2,4], Alpha A. Lee [5], Sarah A. Teichmann [5,6] & Tuomas P. J. Knowles [2,5] ✉

Biomolecular condensates help cells organise their content in space and time. Cells harbour a variety of condensate types with diverse composition and many are likely yet to be discovered. Here, we develop a methodology to predict the composition of biomolecular condensates. We first analyse available proteomics data of cellular condensates and find that the biophysical features that determine protein localisation into condensates differ from known drivers of homotypic phase separation processes, with charge mediated protein-RNA and hydrophobicity mediated protein-protein interactions playing a key role in the former process. We then develop a machine learning model that links protein sequence to its propensity to localise into heteromolecular condensates. We apply the model across the proteome and find many of the top-ranked targets outside the original training data to localise into condensates as confirmed by orthogonal immunohistochemical staining imaging. Finally, we segment the condensation-prone proteome into condensate types based on an overlap with biomolecular interaction profiles to generate a Protein Condensate Atlas. Several condensate clusters within the Atlas closely match the composition of experimentally characterised condensates or regions within them, suggesting that the Atlas can be valuable for identifying additional components within known condensate systems and discovering previously uncharacterised condensates.

For decades, membrane-bound organelles have been recognised as the key mechanism by which eukaryotic cells achieve compartmentalisation. This compartmentalisation allows cells to carry out multiple biological processes simultaneously by creating distinct biochemical environments for each. Biomolecular condensates have been proposed to offer cells an additional layer of spatial organisation that is more dynamic than what membrane-bound organelles can provide[1–5]. To date, numerous biomolecular condensate systems have been identified with several found to regulate key cellular functions, including gene expression, stress response and signal transduction[6–8]. Because of their broad functional roles, condensates have also become promising targets for drug discovery[9–11]. This has sparked significant interest in understanding their composition and the factors that affect it.

Although condensate systems lack membranes, they occur under conditions where demixing is thermodynamically favoured[12,13]. A

[1]Transition Bio Ltd, Cambridge, UK. [2]Yusuf Hamied Department of Chemistry, University of Cambridge, Cambridge CB2 1EW, UK. [3]Department of Chemistry, University of Oxford, Oxford OX1 3TA, UK. [4]Laboratory of Chemical Physics, National Institute of Diabetes and Digestive and Kidney Diseases, National Institutes of Health, Bethesda, MD 20892, USA. [5]Cavendish Laboratory, Department of Physics, University of Cambridge, Cambridge CB3 0HE, UK. [6]Wellcome Sanger Institute, Wellcome Genome Campus, Hinxton, Cambridge, UK. ✉e-mail: ksaar@transitionbio.com; tpjk2@cam.ac.uk

nucleus  cytoskeleton

biomolecular condensates

**Fig. 1 | Cellular biomolecular condensates have complex compositions involving hundreds to thousands of different components.** The number of different condensate types in cells is large with many systems likely yet to be discovered. Experimental techniques used for condensate characterisation either yield sensitive information about a handful of candidate targets (top left) or permit hypothesis-free characterisation without the requirement for pre-defined probes but offer limited resolution in determining protein co-localisation into the same condensate type (bottom right). Here, we combined predictive machine learning models with experimental data from protein interaction and biomolecular condensation studies (top right) to make proteome-wide predictions on the composition of heteromolecular condensates. Image drawn with the aid of BioRender.com.

variety of experimental techniques have, therefore, been developed to characterise their composition. On the one hand, there are immuno-histochemical staining-driven imaging approaches that have helped identify the composition of condensate systems such as stress granules, P-bodies, superenhancers and the nucleolus[14–17]. These approaches can effectively determine the presence or absence of specific proteins in a highly sensitive manner, but their reliance on affinity reagents limits their application to candidate-driven studies without the possibility of discovering hitherto unknown components of condensates (Fig. 1). On the other hand there are a mass spectrometry-based characterisation approaches that extract the condensate system of interest from the cells, typically using an array of purification steps. An elegant interplay of the two approaches is achieved by proximity-labelling methods[18–20], which utilise bait proteins to mark nearby proteins for downstream characterisation, such as mass spectrometry. However, this process still requires prior knowledge of at least one of the components of the condensate of interest, thus, compromising the possibility of discovering previously uncharacterised condensate systems. Gravity-driven fractionation offers an alternative to affinity-based purification, allowing for condensate-agnostic characterisation, but its limited resolution makes it challenging to distinguish between different condensates[21]. It is important to note that even the state-of-the-art methods used for the characterisation of condensates can introduce biases, such as possible changes in "condensate" composition during the purification process.

To complement wet-lab studies, several computational approaches have been developed to study phase separation processes. Most approaches in this space have focused on predicting one-dimensional propensity scores[22–26] but not the composition of heteromolecular condensates. In order to address this challenge, here, we developed a framework that utilised available experimental data to train machine learning models that define the condensation-prone proteome and subsequently combined this information with biomolecular interaction profiles to generate a Protein Condensate Atlas, which predicts the composition of heteromolecular condensates. To identify the condensation-prone part of the proteome, we compared datasets of proteins that undergo phase separation in purified form to those that are found in heteromolecular condensates using NPM1-condensates as an example. We identified significant differences between these two

sets and discovered that high homotypic phase separation propensity is not necessary for protein localisation into condensates. Instead, we found that the composition of heteromolecular condensates is determined by charge and hydrophobicity-mediated protein–protein and protein–RNA interactions. Based on these insights, we constructed a machine learning model that linked protein sequence to its propensity to localise into condensates. When we applied this model to the entire proteome, we identified several proteins with a high likelihood of localising into condensates. Many of the top-ranked targets were later confirmed to localise into condensates by orthogonal immunohistochemical staining imaging. Notably, our analysis showed that the predictive capability extended beyond the NPM1-condensate system. Encouraged by this generalisability, we defined the condensation-prone proteome and used its characterised interactome to generate the Atlas. Several of the predicted condensate systems within this atlas accurately matched the composition of experimentally characterised heteromolecular condensates. This suggests that our framework has the potential to guide the discovery of previously unknown condensate types.

## Results and discussion

### High homotypic phase separation propensity is not a prerequisite for protein localisation into NPM1-condensates

To gain insight into which proteins are present in biomolecular condensates, we analysed the mass spectrometry data from lysate-reconstituted NPM1-condensates that was recently gathered by Freibaum et al. (Supplementary Dataset 1)[17]. NPM1 is an RNA-binding protein that is known to play a central role in the formation of the nucleolus. Purified NPM1 can induce condensation from cell lysate and forms biomolecular condensates that recapitulate the nucleolus. As the dataset only included information on proteins that were enriched into condensates, we combined this information with a mass spectrometry-based proteomics study that had identified a total of 7273 proteins in this same cell line (U2OS; Supplementary Dataset 2)[27]. We found 1008 of these proteins to overlap with the proteins that had been detected in condensates (Fig. 2a, green).

At first glance, it might appear that the remaining 6265 proteins do not partition into condensates (Fig. 2a, red). However, a closer investigation suggested that the proteins that did not partition into

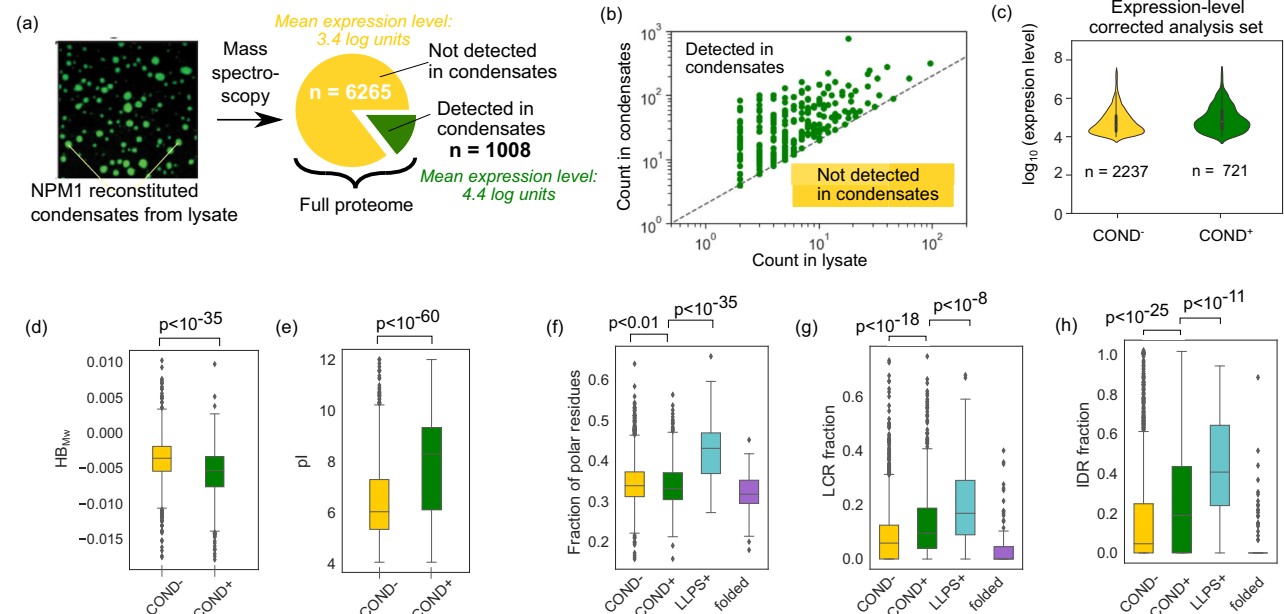

**Fig. 2 | Comparison of the biophysical signatures of proteins that partitioned and did not partition into reconstituted NPM1-condensates. a** When forming reconstituted NPM1-condensates and analysing their composition with mass spectrometry[17], around 1000 proteins from the U2OS proteome as characterised by ref. 27 were found to be enriched into NPM1 reconstituted condensates (green) and the remaining 6265 were not (yellow). The proteins detected in condensates had, on average, tenfold higher expression levels. **b** Comparison of protein count in the condensate and in the lysate indicated that protein concentration was not a key factor defining condensate partitioning. **c** For further comparisons, we created two groups of proteins with similar expression levels (above 3.5 logarithmic units; expression levels were based on the data acquired by ref. 27) but different condensate partitioning propensity, COND+ and COND−. The black rectangle captures the interquartile range (IQR), the shape of the curve describes the estimated probability density of the data and $n$ corresponds the number of proteins in each category. **d** The proteins that partitioned into condensates (COND+; green; 721

proteins) had lower hydrophobicity per molecular weight than those that did not (COND-; yellow; 2237 proteins). **e** The majority of the proteins in the COND+ set had pI values above physiological pH, suggesting they carry a positive charge. **f−h** Comparison of the fraction of polar residues, low complexity (LCR) and intrinsically disordered regions (IDR) between the COND+ (yellow) and the COND− (green) datasets, a set of proteins that undergo homotypic phase separation (LLPS +; cyan; $n = 153$ proteins constructed using the PhaSepDB database[28]) and a set of fully folded proteins very unlikely to undergo homotypic phase separation (purple; 135 proteins)[29]. Across all three features, condensate partitioning proteins (COND+) differed from proteins with a high intrinsic phase separation propensity (LLPS+; two-sided Mann−Whitney $U$-test). In panels **d**−**h**, the centre of the box corresponds to the median, its bounds to the lower and upper quartiles and the whiskers to 1.5 times the IQR from the lower and upper quartiles. Source data are provided in the Source Data file.

condensates also had a lower expression level, with the mean value between the two groups differing by an order of magnitude. To examine if this difference is real (concentration is a key determinant of protein partitioning into condensates) or an assay artefact (measurement is biased to identifying proteins with higher expression levels as characterised by ref. 27), we analysed how the spectral count of each protein in the condensate depended on its concentration. We observed that the protein count in condensate did not depend strongly on its concentration in the lysate, with both highly and lowly expressed proteins showing high counts in the condensate fraction (Fig. 2c). This finding suggested that concentration should not be a key factor underlying partitioning. We thus filtered the full proteome of the U2OS cell line to proteins with expression levels above 3.5 $\log_{10}$ units to create two datasets of proteins with relatively similar expression levels but different condensate partitioning propensity: COND+ and COND−.

Using these datasets, we investigated the biophysical characteristics that drive protein localisation into NPM1-condensates. We noticed that proteins with a high tendency to localise into the condensates (COND+; green) were less hydrophobic ($p < 0.0001$, Mann−Whitney $U$-test; Fig. 2d) and had higher isoelectric points (pI; $p < 0.0001$, Mann−Whitney $U$-test; Fig. 2e) than the set of proteins that did not localise into condensates (COND−; red). The difference in hydrophobicity was present both on the absolute scale and when the values were normalised by molecular weight (Methods). We note that a significant fraction of the proteins that were found in condensates had

isoelectric points above the physiological pH, which leads to their carrying of a positive charge and, thereby, to possible attractive forces to the oligonucleotides likely to be present in these condensate systems. Taken together, these results suggest that both hydrophobicity and charge-mediated interactions contribute to the recruitment of proteins into heteromolecular NPM1-condensates. We replicated the analysis on the reconstituted G3BP1 condensates characterised by ref. 17 Our results showed similar trends, confirming the significance of these findings beyond the NPM1 system (Supplementary Fig. S1a, b).

Next, we aimed to understand if the proteins that were found in condensates share similar properties to those that undergo intrinsic phase separation processes. To this effect, we complemented the datasets above with two additional sets of proteins: a set of $n = 154$ proteins that have been experimentally shown to undergo homotypic phase separation as highlighted in the PhaSepDB database v2.1[28] (Supplementary Dataset 3; Methods) and a set of fully folded proteins very unlikely to undergo homotypic phase separation filtered for a high degree of sequence diversity (Supplementary Dataset 4)[29]. Across these datasets, we compared some key features that have been associated with a high phase separation propensity: the fraction of the protein that is (i) disordered, (ii) of low complexity or (iii) comprises polar residues[30−32]. In all three cases (Fig. 2f−h), we observed the fraction to be significantly lower ($p < 0.0001$; Mann−Whitney $U$-test) for the condensate partitioning set (COND+; green) than for the set that can phase separate homotypically (LLPS+; cyan). This observation suggested that condensate systems likely contain both proteins that

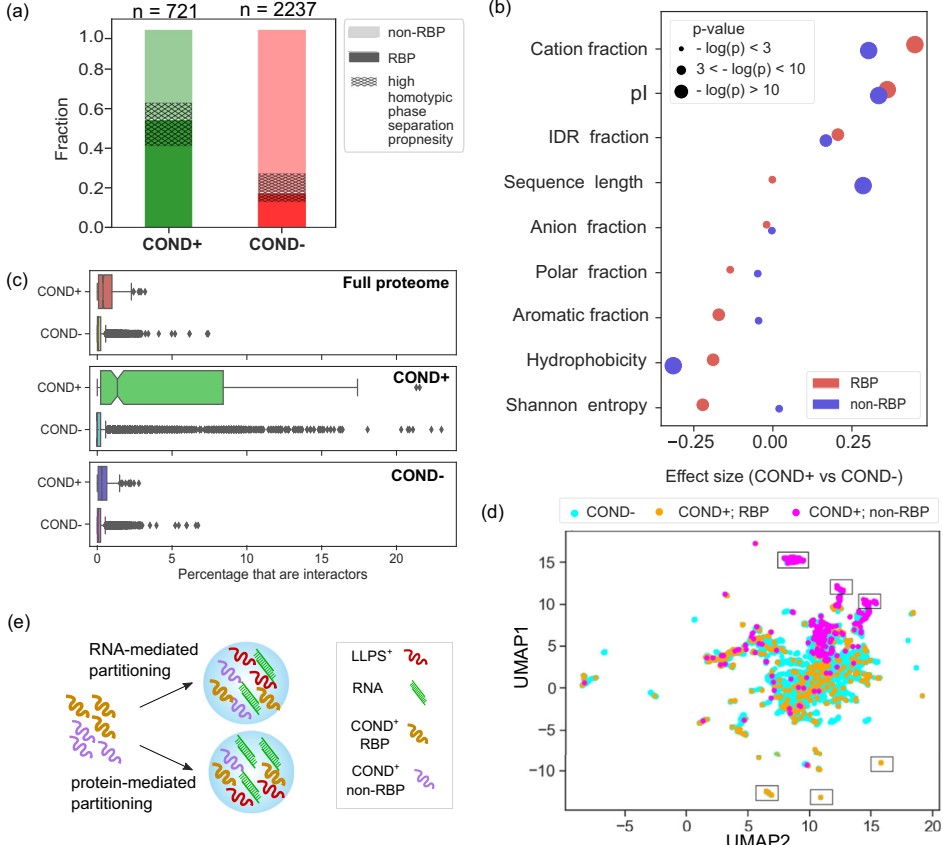

**Fig. 3 | Importance of biomolecular interactions in determining the composition of biomolecular condensates. a** Grouping proteins in the COND+ (green) and COND− (red) datasets according to their RNA-binding character using gene ontology annotations (dark vs. light colour) highlighted that the fraction of RNA-binding proteins (RBPs) was much higher in the COND+ dataset. A notable fraction of the proteins in both sets neither interacted with RNA (dark colour) nor had a high homotypic phase separation propensity (shaded area). **b** When evaluating how proteins in the COND+ and COND− datasets compare when dividing them into RBPs (red) and non-RBPs (blue), for RBPs, we found cationic fraction and pI to be among the key features that governed localisation into condensates (positive effect size indicates the feature has a higher value among condensate partitioning proteins, COND+). Sequence length correlated strongly with localisation into condensate for non-RBPs but it played no role for RBPs. Comparisons were performed using a two-sided Mann–Whitney U-test. **c** When analysing the number of identified

interaction partners (data from StringDB[34]), we observed that proteins within the COND+ dataset had many more interaction partners among proteins that were part of the condensate. Their number of interactors across the full proteome and in the COND− dataset was also larger, but the difference with COND− was less significant. The centre of the box corresponds to the median, its bounds to the lower and upper quartiles and the whiskers to 1.5 times the IQR from the lower and upper quartiles. COND+ and COND− include 721 and 2237 proteins, respectively.
**d** Clustering the proteins based on the similarities in their proteomic interactomes suggested that in some cases, protein recruitment into condensates was mediated by RNA (pink clusters; top) while in other cases the recruitment was driven by protein–protein interactions (orange clusters; bottom). **e** Collectively, these data suggest the importance of both protein- and RNA-interactions in determining the composition of heteromolecular condensates. Source data are provided in the Source Data file.

---

have a high inherent phase separation propensity and proteins that partition into existing condensates through interactions with other components of the condensates. This observation is in agreement with earlier studies that have also found proteins in the nucleolus to have weak enrichment in the features homotypically phase-separating proteins exhibit[33].

## NPM1-condensate composition is determined by both protein–RNA and protein–protein interactions

Having identified that both hydrophobic and charge-mediated interactions contribute to defining the composition of heteromolecular NPM1-condensates (Fig. 2d, e), we aimed to delve deeper into their roles. To this effect, we divided the proteins in the COND+ and COND− datasets into two categories based on whether they were annotated as RNA-binding (GO:0003723). We found that about half of the proteins in the COND+ dataset were RNA interactors as opposed to only 20% of the proteins in the COND− dataset (Fig. 3a). These results strongly indicated that RNA-mediated interactions play a pivotal role in determining the composition of heteromolecular NPM1-condensates.

However, the fact that certain RNA-binding proteins do not localise within the condensates, despite their high expression level, suggests that certain protein–RNA interactions predominate over others. To further understand the role of RNA and RNA-binding proteins (RBPs and non-RBPs, respectively) in condensate formation, we divided both the COND+ and COND- datasets into two groups based on their propensity to bind RNA. We estimated the p value and Cliff effect size for several key biophysical features between the two groups using a two-sided Mann–Whitney U-test. We observed that some features that differed between the COND+ and COND− datasets, such as the abundance of disordered regions, played a comparably important role for RBPs and non-RBPs (Fig. 3b; positive effect size indicates proteins with a high value for the given feature are more likely to partition into condensates). For RBPs (red), partitioning strongly correlated with a high fraction of cationic residues and high pI. For non-RBPs (blue), it also correlated strongly with low hydrophobicity. Sequence length differed distinctly between the two groups with large non-RBPs more likely to partition into condensates; no such size-dependent difference was observed for RBPs.

The finding that close to 40% of the proteins in the COND+ dataset are not RNA-interactors or do no possess a high homotypic phase separation propensity as evaluated by the DeePhase algorithm (Fig. 3a; shaded area; Methods) led us to hypothesise that another feature that may drive protein partitioning into condensates is interactive forces with other proteins that are part of this condensate system. To assess this possibility, we turned to the StringDB database that aggregates protein–protein interaction data from multiple sources to generate protein interaction networks[34]. We counted the number of interactions for each protein in both the COND+ and COND– datasets with proteins that (i) localised into condensates, (ii) did not localise into condensates, and (iii) are found across the human proteome (Methods). We noticed that while proteins in the COND+ dataset had slightly more interaction partners within the COND– dataset and across the proteome (Fig. 3c; top and bottom panels), they had significantly more interaction partners with the COND+ dataset itself (middle panel). This difference highlights the importance of protein–protein interactions in determining the composition of heteromolecular condensates.

Finally, we aimed to obtain a more detailed understanding of the relative roles of protein–protein and protein–RNA interactions in determining the composition of heteromolecular condensates. To accomplish this, we clustered proteins in the COND+ and COND– datasets based on their interaction profiles. Specifically, we performed this step by generating a matrix that described pairwise interactions between all these proteins using data from the StringDB database and by visualising the formed clusters in 2D-space using umap embeddings (Methods)[35]. Additionally, we annotated the points according to whether they were found in condensates and if their gene ontology annotation suggested RNA binding. Our analysis revealed that while some clusters predominantly included RNA-binding proteins (Fig. 3d, orange clusters; GO-term enrichment for the highlighted clusters is shown in Supplementary Dataset 5), there were also clusters that exclusively contained proteins not known to bind RNA (pink clusters; GO-term enrichment for the highlighted clusters is shown in Supplementary Dataset 6). This result further emphasised the importance of both protein–RNA and protein–protein interactions in shaping the composition of biomolecular condensates (Fig. 3e).

## Machine learning models can predict protein partitioning and identify additional components of condensates

In order to comprehensively characterise the localisation of proteins into condensates across the entire proteome, even for proteins with low expression levels, we next set out to develop machine learning models to learn the relationship between protein sequence and its localisation into condensates from available experimental data (Fig. 4a). We created and evaluated four models, each utilising a different strategy for featurising the proteins. The first model used the DeePhase score—a value that has been shown to effectively predict the homotypic phase separation propensity of a protein[29]—as its only input (Fig. 4b; Scaffold; blue). The second strategy used various physicochemical descriptors to describe the proteins (Methods; EngF; red). Given the significance of RNA binding in protein localisation into condensates, the third strategy incorporated this descriptor set with the RNA binding annotation of the proteins (EngF + RNA; green). Finally, the fourth strategy took a hypothesis-free approach to representing proteins. Instead of explicitly defining features, it used a pre-trained SeqVec language model to create protein representations (Methods; SeqVec; cyan)[36] This language model learns fixed-length vectors that capture sequence information in a compressed format, making no assumptions about the important features for the process of interest. We note that with the exception of the third approach (EngF + RNA), all the features can be calculated directly from the protein sequence. All the models were trained on the COND+ and COND– datasets with their hyperparameters tuned through a tenfold cross-validation process described in detail in the Methods section.

We compared the predictive power of each approach by estimating the areas under the receiver-operator characteristic curve (auROC; Fig. 4b) and the precision-recall curve (auPRC; Fig. 4c) on an independent left-out test set. We noticed that the model relying solely on the DeePhase score as the input performed notably worse than all other approaches. As also suggested by earlier analysis (Figs. 2, 3), this was likely due to homotypic phase separation propensity not being the only factor that determines protein localisation into condensates. The remaining four strategies performed comparably well and reached auROC and auPRC values as high as 0.78 and 0.57, respectively. We note that the performance of the models did not get elevated substantially when an explicit feature characterising the RNA-binding character of the sequences was included. One advantage of building models without relying on this feature is the ability to make predictions for every sequence, regardless of RNA-binding annotation availability.

Finally, we deployed the model to assess the probability of each protein in the human proteome to localise into the NPM1-condensates. We observed that the predicted scores were high for the majority of the proteins within the COND+ dataset (Fig. 4, blue distribution) but also for several proteins that had not been seen to localise into the NPM1 reconstituted condensates by Freibaum et al. (green distribution). To verify if any of these proteins are true positives and localise into condensates, we turned to the Human Protein Atlas database that has performed immunohistostaining for a large number of proteins across the human proteome using U2OS as one of its model cell lines[37]. We visually inspected the images of the top 10 highest and lowest-scoring proteins filtered for proteins not present in the training data. We observed the former set to be enriched in proteins that localise into condensates (Fig. 4d top row; four out of top 10 targets showed clear condensates, a few additional with less well-defined condensates as has been summarised in Supplementary Dataset 7) relative to the latter set (bottom row). The images further suggested that these proteins localised into the nucleus and could be part of the nucleolus, of which NPM1 is a key component[38]. Taken together, these results clearly demonstrate the ability of the model to extend beyond the sequence space covered in the training set.

## Predictive capability extends beyond NPM1-condensates and permits the construction of Protein Condensate Atlas

Although the model developed in the previous section was trained on data that characterised the composition of reconstituted NPM1-condensates, the input features used were not specific to this particular condensate system. We hypothesised that since the formation of biomolecular condensates is thought to be driven by common interactions, such as weak multivalent interactions between disordered and folded regions[39,40], the model might assign high scores to proteins that localise into other condensate systems. To test this hypothesis, we turned to the PhaSepDB database[28], which aggregates data from various publications characterising membraneless organelles (MLOs). Since our predicted condensate localisation scores peaked at a relatively low value (Fig. 4a), we converted them to normalised scores based on their percentiles in the distribution. We focused on the six largest condensate systems for human proteins within the PhaSepDB database: nucleolus, stress granules, P-bodies, nuclear speckles, paraspeckles and spliceosomes and examined the predictions of our model on the proteins that have been characterised to be part of these systems. We found that the predictions for the components of all these six condensate systems were higher than for proteins without any condensate-associated annotations (Fig. 5a). We further confirmed this finding by comparing the 50th, 75th and 90th percentile values for each of the distributions using bootstrapping to estimate the uncertainties in each value (Supplementary Fig. S2a–c).

Motivated by this observation, we set out to explore if it is possible to partition our model-predicted condensation-prone proteome into individual condensate systems by utilising previously

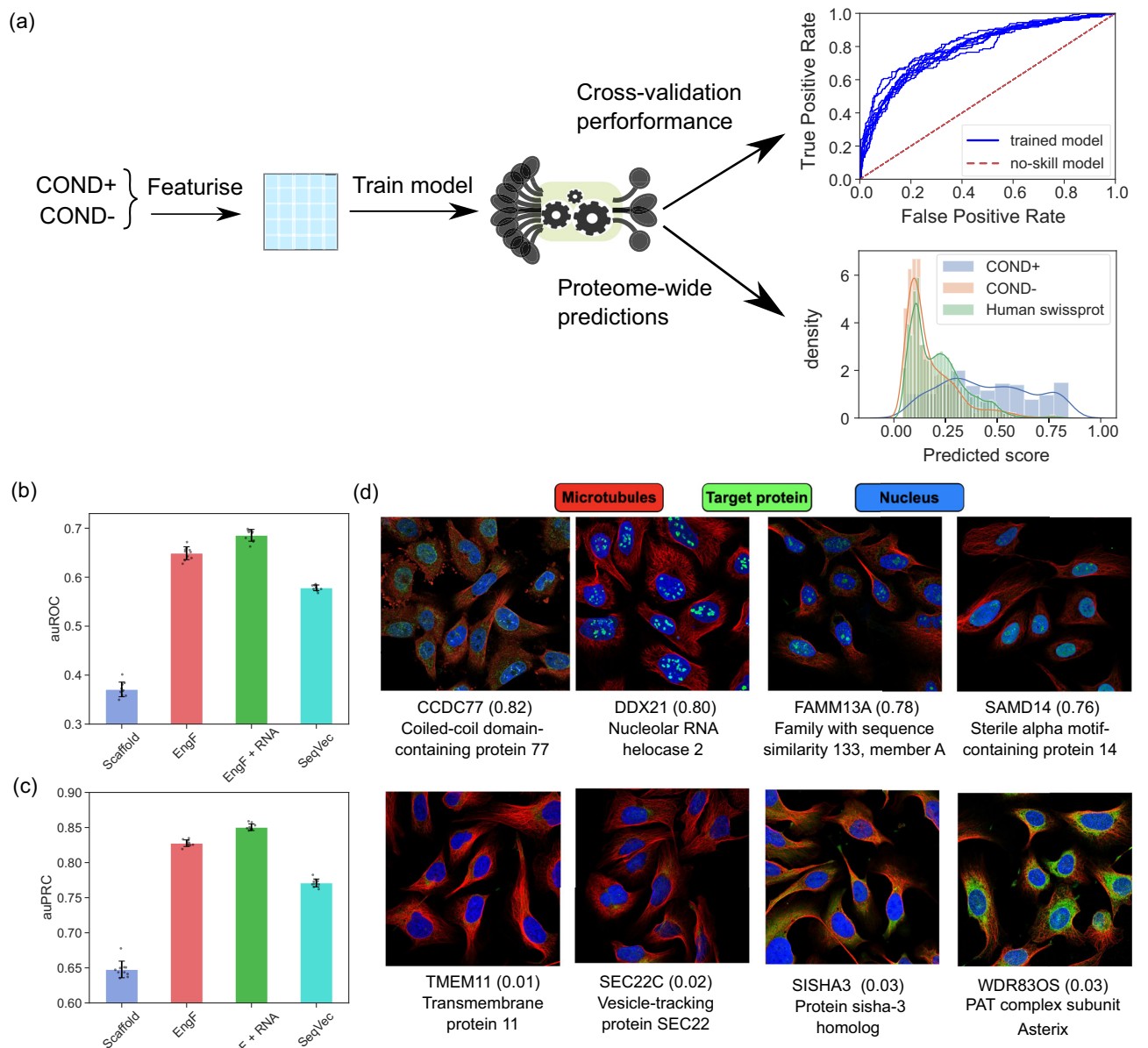

**Fig. 4 | Proteome-wide predictions of protein partitioning into NPM1-condensates. a** A machine learning model was trained to distinguish between the proteins in the COND⁺ and COND⁻ datasets using a cross-validation strategy to find optimal model architecture and hyperparameters (top). The model was then deployed to evaluate the propensity of all the proteins across the human proteome to localise into condensates (bottom). The blue and orange distributions correspond to predictions made on the training data and the green distribution to the predictions made on the remainder of the human proteome. Panel drawn with the aid of BioRender.com. **b** Area under the receiver-operator characteristic curve (auROC) using four different featurisation strategies: homotypic phase separation propensity score as quantified by a single value[29] (blue; Scaffold); sequence-derived features (red; EngF), sequence-derived features in combination with RNA-binding annotation (green; EngF + RNA), SeqVec embedding (cyan). **c** Performance of the models as quantified by the area under the precision-recall curve using the minority class (proteins that are partitioned into condensates). Data on panels **b**, **c** are presented as the mean value ± standard deviation with the values calculated over 10 bootstrapped estimates. Individual values are shown as circles. **d** Examples of proteins that were not part of the COND⁺ or COND⁻ datasets (panel (**a**), green distribution) and were predicted to have high (top) or low (bottom) scores. The images were acquired in the U2OS cell line as part of the Human Protein Atlas project[37]. The immunofluorescence profiles suggest that many of the highly-scoring proteins are likely to form condensates in U2OS cells in contrast to the low-scoring proteins. Source data are provided in the Source Data file.

characterised biomolecular interaction profiles to identify which proteins co-localise into the same condensate system. The development of a capability to predict the composition of heteromolecular condensates would be a qualitative advancement over approaches published to date, which have focused on predicting homotypic phase separation propensity or protein localisation into condensates[22,23]. To create such a Protein Condensate Atlas, we first defined the predicted condensate localising proteome as proteins with a high intrinsic phase separation propensity (DeePhase score above 0.75) or a predicted partitioning score above 0.25 (Fig. 4a, green line; the optimal threshold was determined using the Youden J-statistic). We then integrated information on biomolecular interactions from the StringDB database and consensus clustered the interaction profiles of the human proteome (Methods; Fig. 5b). This process yielded a total of 133 clusters. For 62 of these, our models predicted that at least half of the proteins would localise into condensates, suggesting that they may correspond

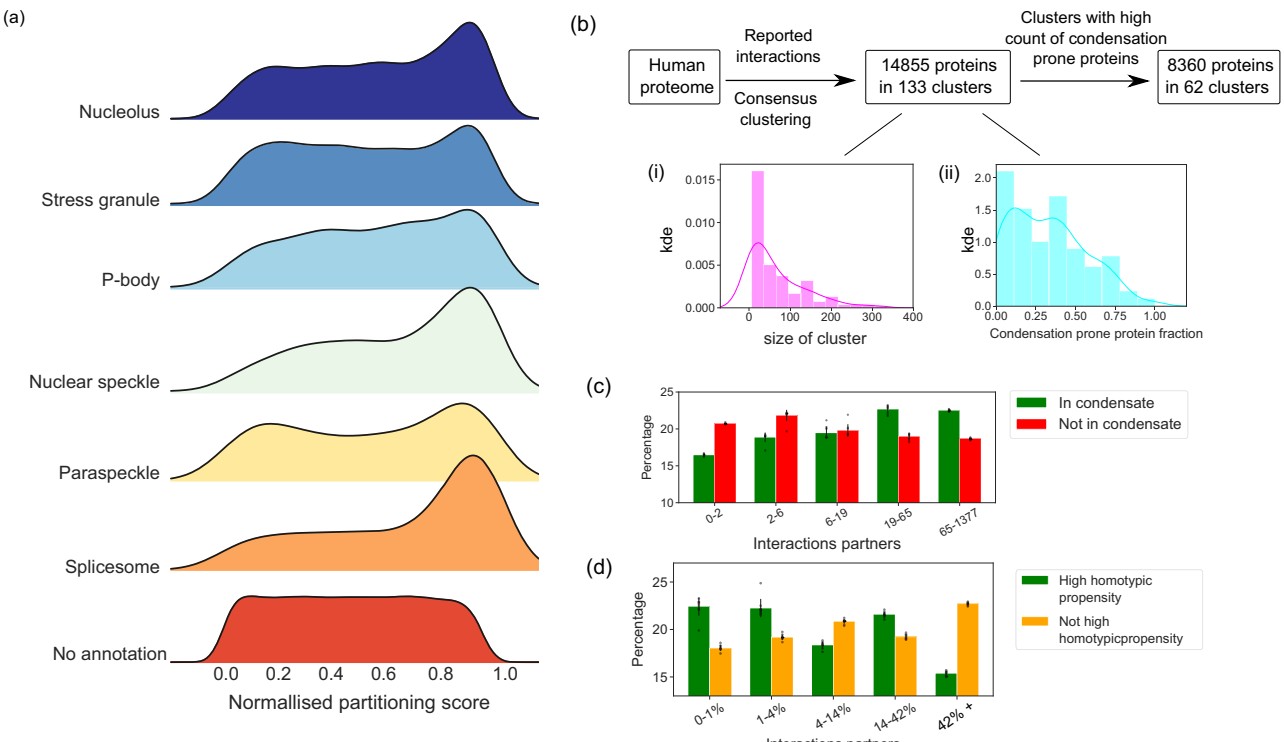

**Fig. 5 | Construction of the Protein Condensate Atlas. a** The distributions of the normalised condensate localisation scores for proteins that have been experimentally observed in the nucleolus (top row; data from PhaseSepDB[28]) but also in other characterised condensate systems (middle five distributions) are higher than the scores for the proteins with no condensate annotation (bottom distribution). **b** Consensus clustering of proteins based on their interaction profiles in StringDB[34] followed by filtering of the clusters by condensation propensity resulted in the prediction of over 60 condensate clusters (Supplementary Dataset 8). **c** Proteins predicted to be within the condensate clusters (green) had more interaction partners than proteins not predicted to localise into condensates (red). The y-axis indicates the fraction of proteins in each category. **(d)** Within each predicted condensate cluster, proteins with a high homotypic phase separation propensity had fewer interaction partners than condensate localising proteins without a high predicted homotypic phase separation propensity. Data in panels **c**, **d** are presented as the mean value ± standard deviation with the values calculated over ten bootstrapped estimates. Individual values are shown as circles. kde stands for kernel density estimate. Sorce data are provided in Source Data file.

to condensate systems. The composition of the predicted condensate clusters, which we refer to as our predicted Protein Condensate Atlas, can be found in Supplementary Dataset 8. We note that our algorithm predicts each protein to be located in only a single cluster.

Next, we set out to examine how biomolecular interactions and condensation propensity, which are the two key inputs into the model, correlate with the predictions of the Atlas. To this effect, we first quantified the number of confirmed interaction partners that the proteins within and outside of the condensate clusters had, utilising the data reported in StringDB. We observed that proteins not part of the condensate clusters (Fig. 5c, green) tended to have a larger number of interaction partners compared to proteins not predicted to localise into condensates (red). This trend aligns with our finding when we characterised the composition of NPM1-condensates (Fig. 3c).

We additionally analysed interactions within the predicted condensate clusters. Specifically, we used data from StringDB to count the number of interactions that each protein within a predicted condensate cluster can form with other proteins in the same cluster. The interactions within an exemplary condensate system are shown in Supplementary Fig. S3a, with the distributions of the interaction counts highlighted in Supplementary Fig. S3b. For this particular cluster, we found no significant difference in the number of interactions between condensate localising proteins with and without a high predicted homotypic phase separation propensity (Supplementary Fig. S3b, orange and green, respectively). However, when we performed the analysis globally across all the predicted condensate clusters, we observed that proteins without a high homotypic phase separation propensity tended to have a higher number of interaction

partners (Fig. 5d; the y-axis values are normalised for condensate cluster size to allow comparison between clusters of different sizes) than proteins that had a high phase separation propensity. This observation further highlights the key role that heteromolecular interactions play in protein recruitment into condensates (Fig. 3e) and its distinction from homotypic phase separation processes.

**Evaluating the predictions of the Protein Condensate Atlas**

Finally, we sought to validate the predictions of our Protein Condensate Atlas. To the best of our knowledge, this is the first atlas of its kind, and the task lacks an established benchmark. Nonetheless, we conducted several analyses to gain insight into the performance and limitations of the Atlas by evaluating how well the predictions capture the composition of previously studied condensates.

To begin, we examined whether the predicted condensate clusters were enriched for proteins that have experimentally been shown to co-localise within the same condensate using the PhaSepDB database[28]. This database collates data on a variety of membraneless organelles (MLOs) and, importantly, provides an independent test set from the input data that was used to construct the Atlas. We note that the data in the PhaSepDB is non-exhaustive, and information on the composition of other condensates may have been published separately. We focused on human MLOs that had at least 10 detected proteins (a total of 14 systems). We used the data in the PhaSepDB as provided for all the MLOs with the exception of the stress granule, where we also narrowed the full set of 1536 proteins down to 140 proteins that have been reported to form the core of the stress granule (Supplementary Dataset 9)[41] that is the more stable part of this MLO[42].

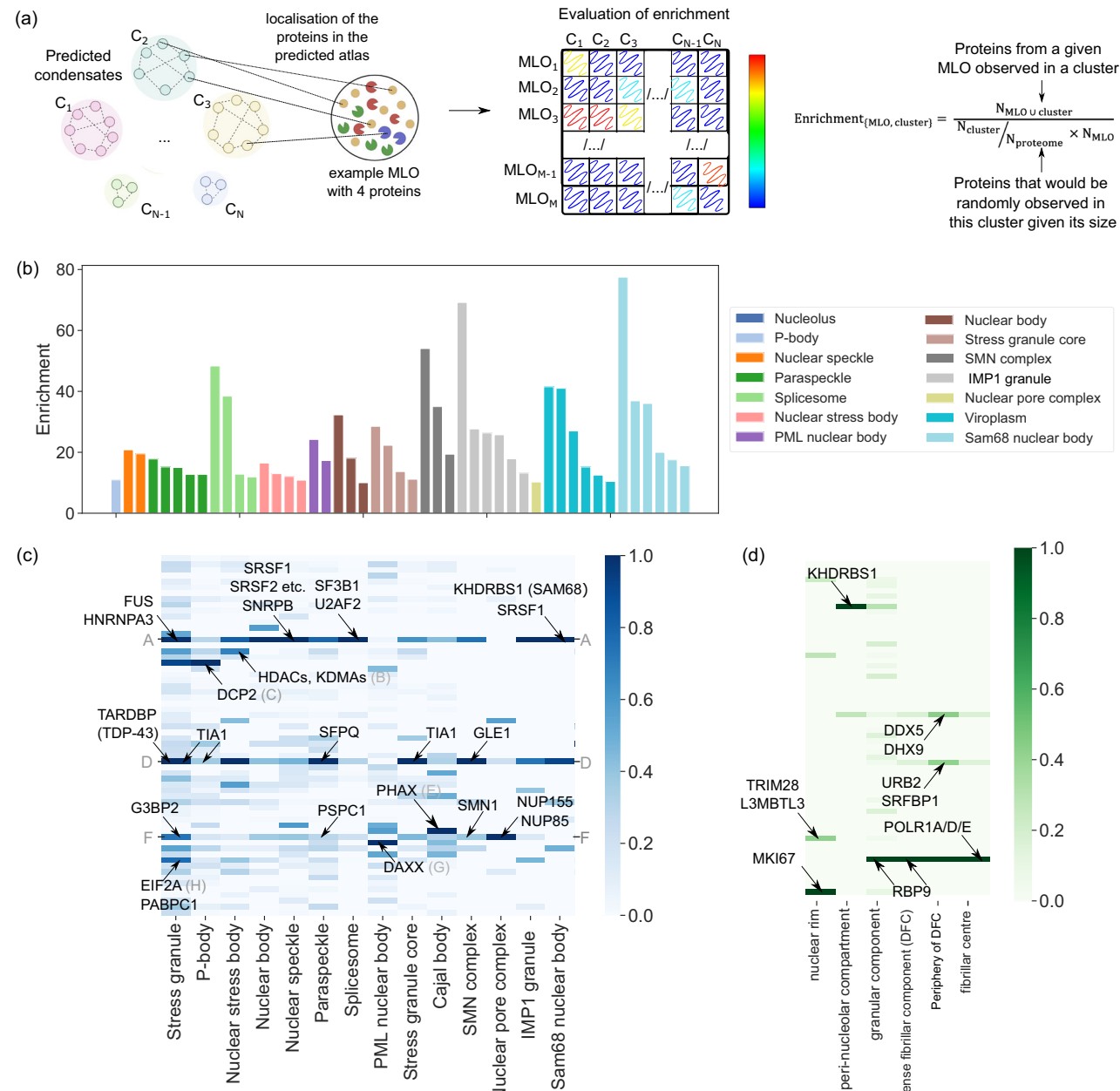

**Fig. 6 | Validation of the results of the predicted Atlas. a** The composition of each predicted condensate cluster was compared to characterised MLOs by estimating the enrichment of the proteins within each cluster relative to random sampling (Methods). **b** Enrichment values of the predicted clusters for 14 MLOs described in PhaSepDB. Only values above 10 are shown. For most systems, enrichment values over 10 are observed and they are concentrated to one or a handful of the predicted condensate clusters, highlighting that our Atlas captures the composition of previously characterised condensate systems. **c** Enrichment values for the MLOs whose

composition has been characterised highlight key clusters into which the components of the condensates are enriched. Enriched clusters include proteins that are known markers for the specific condensate system (highlighted in black) as well as those that have not been detected in these condensates, suggesting additional candidate proteins for these MLOs (full list in Supplementary Dataset 8; letters A-H annotate different clusters). **d** Same analysis for different regions of the nucleolus, highlighting that the Atlas is able to capture sub-MLO level organisation. Source data are provided in Source Data file.

To determine the enrichment values for each of these 14 systems for all the 62 clusters in our Atlas, we counted the number of proteins within each predicted cluster that had been found in a specific MLO and compared this value to random sampling, as exemplified in Fig. 6a and in the Methods section. The enrichment values for all predicted condensate clusters for each of the MLOs are summarised in Supplementary Dataset 10. Moreover, all the enrichment values of 10 or above are highlighted in Fig. 6b. Notably, with the exception of the nucleolus, for all the MLOs for which experimental data was available in the PhaseSepDB, we observed a minimum of tenfold enrichment of their components into one of the predicted condensate clusters. In several

cases (e.g. Sam68 nuclear body, IMP1 ribonucleoprotein granule, SMN complex and spliceosome), the enrichment values exceeded 40. These high values indicated that the predicted condensate clusters in our Atlas closely resemble experimentally observed compositions of MLOs. Additionally, for each condensate system, the Atlas predicted additional proteins that may be present in these MLOs, as demonstrated in Supplementary Fig. S4. The full list of the additional predicted components can be found in Supplementary Dataset 8.

We next examined the relationships between the predicted clusters and known condensate types in more detail. Since the calculated enrichment values varied greatly between MLOs, we first normalised the

values to the highest observed enrichment score for that particular MLO. The result is shown in Fig. 6c. It is clear that some of the clusters resemble multiple MLOs (e.g. clusters A, D, F), which is in agreement with the experimental observations[28]. These clusters include proteins that are involved in a variety of cellular processes and can thereby be located in several condensate systems (FUS and NHRNPA3 in stress granules, nuclear speckles and paraspeckles; TIA1 in stress granules and P-bodies). The results additionally highlight that most MLOs exhibit some uniqueness in their predicted composition. Their known components, including key markers, are predicted to be located in clusters that are specific to this MLO (G3BP1/2 and PABPC1 in stress granules; DAXX in PML nuclear bodies, NUP family proteins in nuclear pore complex).

As mentioned, the nucleolus stood out for not showing a high enrichment for any of the predicted clusters (Fig. 6b and Supplementary Dataset 10). We hypothesised that this low enrichment could be due to its large size and sub-condensate organisation. Indeed, the largest condensate cluster in our predicted Atlas included 400 proteins, while the nucleolus, as reported in the PhaSepDB has over 1000 observed components. The nucleolus is known to be a complex system with distinct regions, and its composition has recently been characterised in detail by ref. 43, who identified six distinct regions within it (Supplementary Dataset 11). Using a similar protocol as described above, we calculated the enrichment values for all the clusters with respect to the characterised subregions within the nucleolus and found the maximum enrichment values to be 50 or above for most of the regions (Supplementary Fig. S5). Moreover, the enrichment matrix showed clear sparks for all the regions (Fig. 6d), indicating the different regions of the nucleolus are indeed enriched into distinct clusters within our predicted Atlas. We note that the predictions are able to best distinguish the nuclear rim and the per-nucleolar compartment, but the fibrillar regions (fibrillar centre and dense fibrillar component) and the granular component are poorly separated. Nevertheless, this finding suggests that our predicted Atlas shows the potential to identify local structures within condensates.

In summary, our validation analysis highlights that while the data in the StringDB database is constructed without any emphasis on whether interactions occur below or above saturation concentrations, in combination with condensation propensity prediction models, these data can be used to predict the composition of biomolecular condensates. Furthermore, our observations indicate that approximately a quarter and a half (18 and 28 out of 62) of the predicted condensate clusters within the Atlas show at least a tenfold or a fivefold enrichment for one of the 14 experimentally characterised MLOs or their subregions, respectively. This means that the remaining clusters in our predicted Atlas (Supplementary Dataset 8) have the potential to describe the components of condensates whose compositions have not yet been characterised in databases. Finally, we note that the accuracy of our predicted Atlas for capturing previously characterised MLOs could be further optimised by additional parameter turning (e.g. the number of cluster centres in the clustering step, thresholds used for defining the condensate localising proteome or extracting clusters that correspond to condensates). However, we intentionally focused on developing an unsupervised algorithm rather than a supervised machine learning task, as this ensures that the algorithm does not become overly tailored to the composition of these few MLOs that have already been characterised.

In summary, using data from mass spectrometry-based characterisation of NPM1-condensates, we analysed the factors that determine protein localisation into these condensates. We found that the features known to influence homotypic phase separation processes, such as protein disorder and low sequence complexity, play a significantly smaller role in determining the composition of heteromolecular condensates. Instead, we found protein partitioning into condensates to be driven by charge and hydrophobicity-mediated biomolecular interactions with both protein–RNA and protein–protein interactions defining the final composition. Based on these findings,

we developed a machine learning model that linked protein sequence to its propensity to localise into heteromolecular condensates. The model showed good accuracy within the available data, and was able to identify additional condensate components, as confirmed by independent experimental validation. This demonstrates the potential of computational approaches in describing the full proteomic content of condensate systems, even in cases of low expression that may not be easily studied experimentally. Finally, we combined our machine learning model with proteomic interaction profiles to build a Protein Condensate Atlas. This atlas revealed key clusters that aligned with known membraneless organelles and condensate systems, indicating that our approach can guide the identification of additional components within established condensate systems and aid in the discovery of uncharacterised condensate types.

## Methods

### Preparation of datasets

Partitioning of proteins into reconstituted NPM1-condensates was evaluated based on mass spectrometry measurements performed by Freibaum et al.[17]. Specifically, we focused on proteins for which the study recorded a partitioning coefficient above two (Supplementary Dataset 1). All the proteins that had been characterised to be present in U2OS cells (Supplementary Dataset 2) but were not found to have a partitioning coefficient of at least two were considered not to partition into condensates. We then removed all proteins whose GO-terms (biological process, molecular function of cellular component) included the keyword "mitochondria", as these proteins were likely observed in the condensate fraction as a result of undesired co-purification[17]. As described in the Main Text, the positive and the negative datasets were filtered down to proteins that had their expression levels above 3.5 log-units to create two datasets with comparable expression levels but different condensate partitioning characters. The datasets were referred to as COND+ and COND−, respectively. These datasets were compared and contrasted to the LLPS+ dataset (proteins with 'PS-self' annotation in the PhaseSepDB database[28]; Supplementary Dataset 3) and the dataset "folded", which was designed to include a sequence diverse set of fully folded proteins[29] (Supplementary Dataset 4).

### Biophysical and biomolecular interaction-related feature sets

A range of biophysical features were calculated to characterise the protein sequences. This included the isoelectric point (pI; calculated using the Python package BioPython[44]), the hydrophobicity (evaluated as the sum of the individual hydrophobicity values of the amino acids in the sequences based on the Kyte and Doolittle hydropathy scale[45]), the fraction of the low-complexity region (LCR) for each sequence (estimated using the SEG algorithm with standard parameters[46]) and the fraction of residues that were part of disordered regions (IDR; estimated using the IUPred2a algorithm[47]).

The following features were included in the hand-crafted feature set (EngF) when building the machine learning models: sequence length, a fraction of the sequence that was part of the LCR, sequence hydrophobicity, Shannon entropy[29], isoelectric point (pI), the fraction of the sequence that was part of the IDR, the delta parameter that describes the patterning of the hydrophobic residues as described in the CIDER package[48], the count of each residue in the sequence and in its low-complexity regions, and the fraction and the count of the different types of residues (hydrophobic, aromatic, cationic, anionic) in the sequence and the low-complexity regions. When estimating the fractions of different types of residues, the amino acids were grouped as follows: hydrophobic residues—alanine, isoleucine, leucine, methionine, phenylalanine and valine; aromatic residues—tryptophan, tyrosine and phenylalanine; cationic residues—lysine, arginine and histidine; anionic residues—aspartic acid and glutamic acid.

A sequence was considered RNA-binding if its GO-annotation[49] included the term GO:0003723. The protein-based interactors of each

protein were extracted from the StringDB database. Two proteins were considered to interact when the confidence score for the interaction was above 700.

## Protein sequence embeddings

Protein language model-derived embeddings were evaluated from the protein sequences using the SeqVec algorithm[36]. Specifically, three-dimensional embedding vectors ($1024 \times$ sequence length $\times 3$) were calculated for each sequence using the pre-trained SeqVec model. The dimensionality of the vectors was reduced by averaging the vectors across all the residues and then summing the three 1024-dimensional vectors−a protocol that has been shown to work effectively when predicting protein properties via transfer learning[36]. This process resulted in 1024-dimensional embedding for each sequence that served as the input to the machine learning algorithms.

## NPM1 condensate localisation model training and performance evaluation

Five different models were evaluated, each utilising a different approach to protein featurisation as described in more detail in the Main Text and outlined in Fig. 4b and in the Methods sections above. All models were based on random forest architecture with the following hyperparameters ranges were considered: $n_{estimators}$: {20, 50, 100, 200}, max_depth: {3, 5, 7, 10}, min_samples_leaf: {1, 2, 4}, max_features: {sqrt, auto}. 20% of the data was set aside as an independent test set in a stratified manner, keeping the ratio of the two classes (COND+ and COND−) fixed. On the remaining 80% of the data, the optimal set of hyperparameters was determined in a tenfold cross-validation process by requiring the area of the receiver-operator characteristic curve (auROC) to be at its maximum. Within each fold, 80:20 split between the train and the validation data was used. The split was similarly performed in a stratified manner so that the ratio of the two classes would be conserved. All classifiers were built using the Python scikit-learn package[50]. A random seed was set to 42 throughout the analysis.

## Construction of the Protein Condensate Atlas

To cluster the proteins according to their interaction profiles, we used the data in the StringDB database[34] and estimated the N × N correlation coefficient matrix between all the proteins of interest, where N corresponds to the total number of proteins analysed. Following this calculation, we performed k-means clustering on the interaction matrix. The clustering was run twice and with $n = 50$ cluster centres. The process was performed using the Python scikit-learn package[50] with two different seeds. Confidence clusters were then extracted by considering two proteins to be part of the same cluster if they clustered together on both occasions. The rest of the details around the Atlas construction are outlined in the Main Text. When binary characterisation of the StingDB data was required (e.g. when counting interactions), a confidence threshold of 0.7 was used to filter the data down to confident interactions, as is suggested in the database manual.

## Evaluation of the predictions of the Protein Condensate Atlas

The composition of every cluster was compared to every MLO by calculating the enrichment factor, $ENR_{cluster-MLO}$, for each cluster−MLO pair. The enrichment factor for each pair was defined as the ratio between the proteins in that cluster that had been experimentally observed to be part of this MLO (using data collated in the PhaSepDB database) and the number of proteins in that cluster that would be part of this MLO if the proteins had been allocated to clusters randomly:

$$\text{Enrichment}_{cluster-MLO} = \frac{O(\text{cluster})}{E(\text{cluster})} = \frac{\text{cluster} \cup \text{MLO}}{p_{cluster} \times N_{MLO}} \quad (1)$$

where $O(\text{cluster})$ and $E(\text{cluster})$ are the observed and the randomly expected protein counts in the cluster for this MLO, respectively, $\cup$ is the union of the two groups, $p_{cluster}$ is the probability that a randomly chosen protein is part of this MLO and $N_{MLO}$ is the number of proteins that belong to this MLO.

## Reporting summary

Further information on research design is available in the Nature Portfolio Reporting Summary linked to this article.

## Data availability

All the data associated with the manuscript is available from Zenodo under the accession code/ https://doi.org/10.5281/zenodo.10844391. (https://zenodo.org/records/10844392). Source data are provided with this paper.

## Code availability

All the code associated with the manuscript is available from www.github.com/kadiliissaar/ProteinCondensateAtlas[51].

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

## Acknowledgements

We would like to acknowledge the Schmidt Science Fellowship in partnership with the Rhodes Trust (K.L.S.), St. John's College Research Fellowship (K.L.S.), the National Institutes of Health Oxford-Cambridge Scholars Programme (L.L.G.), the Cambridge Trust's Cambridge International Scholarship (L.L.G.), the Intramural Research Programme of the National Institute of Diabetes and Digestive and Kidney Diseases at the National Institutes of Health (L.L.G.), and the European Research Council (T.P.J.K.). The authors gratefully acknowledge funding from the European Research Council under the European Union's Horizon 2020 research and innovation program through the ERC grant DiProPhys (agreement ID 101001615).

## Author contributions

K.L.S. and T.P.J.K. conceptualised the study. K.L.S., R.M.S. and K.B. analysed the data and built the models, A.S.M., L.L.G., A.A.L. and S.A.T. contributed analysis tools. K.L.S. wrote the original draft, all authors reviewed and edited it.

## Competing interests

T.P.J.K. and S.A.T. are co-founders of Transition Bio and K.L.S., R.M.S., K.B., S.A.T. and T.P.J.K. are current or previous consultants or employees. The remaining authors declare no competing interests.
