## [Peer Review File · Nature Communications]

Reviewers' Comments:

Reviewer #1:

Remarks to the Author:

The authors developed a methodology for evaluating the compositional landscape of biomolecular condensates. They divided the condensation-prone proteome into condensate types based on an overlap with previously determined biomolecular interaction profiles to generate a Protein Condensate Atlas. There are several problems:

1. As an atlas resource, the authors need to build a webserver of the resource. So, the users can search, explore, plot, and download the data.
2. The authors need to upload the training/test data and the model codes onto GitHub or other publicly available websites.
3. Are the biomolecular condensates stable? In other words, in which condition, the authors performed the measurement? Will the results be the same when they are measured in other conditions? If they are not stable, there is little meaning to do so. Have the authors tested on various conditions?
4. There are several phase separation prediction models, such as PMID: 37374089. The authors need to make comparisons or discussions of how their model is different and better.
5. Have the authors evaluated the prediction results with experiments? It seems that the authors did not do any validation during the construction of the Protein Condensate Atlas. They assumed the prediction model can make all the classifications.
6. With the benchmark dataset, it is easy to build such prediction models. Therefore, the most important contribution was the dataset. The predicted Protein Condensate Atlas should mark the benchmark and predict data. They have different quality levels.

Reviewer #2:

Remarks to the Author:

The manuscript describes use of mass spectrometry data for components of an NPM1 (nucleolus-like) condensate to develop a predictor of the components of biomolecular condensates. This is an important goal, given the increasing understanding of the importance of condensates in regulating and organizing biology, and given the experiment challenges in defining components of condensates. The authors utilize machine learning approaches initially to train a nucleolus-specific condensate predictor. They found it was not nucleolus-specific, and the authors supplemented the predictor with protein-protein interaction data from stringDB to add localization/condensate-specificity. Then they applied it to the proteome and referred to the results as a Protein Condensate Atlas. The mass spec data from using NPM1 condensates to generate nucleoli-like condensates by enriching proteins from U2OS cell lysates (Freibaum, B. D., Messing, J., Yang, P., Kim, H. J. & Taylor, J. P. High-fidelity reconstitution of stress granules and nucleoli in mammalian cellular lysate. *J. Cell Biol.* 220, 2021) is a powerful dataset and the authors of this manuscript have used machine learning approaches to extract information about cellular condensates from these mass spec data. They found some interesting general principles regarding the distinction between proteins found to have a high propensity for homotypic phase separation and those proteins found to localize to the NPM1 condensate, some of which are specific to RNA- (or DNA-) containing condensates, such as positive charge and RNA-binding annotation. While the results are unfortunately not clearly interpretable in terms of known components of biomolecular condensates and there is some lack of precision in error analysis and overstatement of results, the approach is intriguing and the conceptual insights and datasets are of value towards addressing the goal of defining the components of biomolecular condensates, an important developing area.

Specific comments:

Line 29: The power of proximity labeling approaches is not appropriately recognized here. With large numbers of baits, this approach can be extremely valuable. Of course, as the authors note, it is still candidate driven.

Lines 29-35: The problems with characterizing condensates which the authors describe miss the primary concern with any "purification" of condensates changing their composition or dissolving

them.

The approach driven by NPM1-condensates is biased to nucleic acid-containing condensates. Much of the text needs to be changed to be more specific to NPM1-condensates (as correctly done on lines 152 and 164, 167). For example, line 53 "into condensates" should be "into NPM1-condensates", line 71 "localisation into condensates" should be "localisation into NPM1-condensates", line 79 (and 92/93, and 97) "heteromolecular condensates" should be "heteromolecular NPM1-condensates" and line 91 "Condensate composition" should be "NPM1-condensate composition".

Line 65 and Fig 2(c) (and Fig 2(a)). It is assumed that expression level is defined based on the Beck et al mass spec approach referenced in Fig 2(a). This should be clarified in text rather than just with a reference in Fig 2(a).

Figure 4. The fact that there is no absolute negative set means that the validation measures are challenging. In addition, components can belong to multiple condensates. This needs to be discussed. The ROC and PRC curves should be shown in supplementary data. In general, there does not appear to be a serious evaluation of the error rate of their predictor (and therefore the atlas).

The cross-validation procedure with the ROC analysis appears to be biased. The authors wrote (lines 247-250): "The optimal set of hyperparameters for each model was determined in a 10-fold cross-validation process by requiring the area of the receiver operator characteristic curve (auROC) to be at its maximum. Within each fold, 80:20 split between the train and the validation data was used." If the authors used the validation split to optimize hyperparameters, the auROCs estimated on those validation splits are going to be over-estimated. To report unbiased auROCs, they need to split off unseen "test" data that is not used in either parameter or hyperparameter optimization.

The results in Figure 5(a) are not strong with most of the proteins annotated in the various condensates having a prediction around 0.2, near the peak of the curve for proteins with no annotation. The authors do not acknowledge this. Given this, what is the meaning of the predictor score?

Lines 181/182: "This process resulted in the generation of over 230 clusters. Around 50 of these had at least half of the proteins in the cluster condensation-prone, suggesting they may constitute condensates." The authors should consider the possibility that other clusters besides the 50 listed in supplementary dataset 7 could represent condensates and should also list the components within all 230 clusters in a supplementary dataset.

Fig 5(c) is very unclear. The message (lines 185/186: "Remarkably, we found two thirds of the clusters to show over two-fold enrichment in proteins that have been experimentally seen to co-localise into the same condensate system") is not supported since it is not obvious how the comparison is being made. What are these plots and how are they generated? Also, why is this enrichment "remarkable" considering that the protein-protein information from stringDB was included in making these clusters. Since proteins are more likely to interact with other co-localized proteins, wouldn't this observation be expected?

The claim to have generated a protein condensate atlas is not really supported. There is no clear list of clusters that are supposed to actually match specific described condensates. The list in Supplementary dataset 7 is not annotated with suggested biomolecular condensates "matching" each or potentially included within the clusters given.

The authors need to acknowledge that specific protein components are often part of more than one biomolecular condensate. Some obvious examples are proteins that have some degree of nuclear localization and some cytoplasmic localization, and are found in distinct condensates in these environments. Other examples include proteins that are found in highly related RNA "granules" including P bodies, stress granules and mRNA transport granules.

Minor points:

Multiple figures used "density" as a y-axis label without clear definition of what this means. Density is likely dependent on the bin size for the x-axis values but it is hard to match these numbers to the total values being plotted.

Typographical errors

Line 119 "suggested the bind RNA" should be "suggested they bind RNA"

Line 158 "they may part of condensates" should be "they may be part of condensates"

Line 168 "is through to be driven by universal forces" should be "is thought to be driven by universal forces"

Line 166 "Although the model" should be "Although the model"

Line 174 "scores for these proteins that did not have any" should be "scores for those proteins that did not have any"

Fig 3(a) legend: "phase separation propnesity" should be "phase separation propensity"

...

Reviewer #3:

Remarks to the Author:

The article by Saar, et al., addresses relationships between protein sequences and associated physicochemical features, and protein localization within cellular biomolecular condensates. The authors utilized a previously published protein condensates list (COND+ proteins) generated using the nucleolar protein, NPM1, and U2OS cell lysate. The authors augmented this list with a COND- list generated using other existing data on all proteins in U2OS cells. The authors showed that protein partitioning into the condensate fraction was independent of protein concentration in lysate, and that some physicochemical properties of proteins were enriched or depleted in the COND+ group. They also noted that COND+ proteins were less enriched in phase separation-associated features than proteins known to undergo homotypic phase separation, with their interpretation being that condensate partitioning and autonomous phase separation are driven by different sets of protein features. The authors next showed that COND+ proteins often bind RNA, and that the physical properties of RNA-binding and non-RNA-binding COND+ proteins are somewhat different. Using StringDB, the authors showed that COND+ proteins experience larger numbers of interactions than COND- proteins, and are most likely to interact with other COND+ proteins. At this point in the ms, the authors propose that proteins partition into condensates either through protein-RNA or protein-protein interactions. The authors next developed machine learning (ML) models to predict condensation behavior using various combinations of feature- and sequence-based properties of COND+ and COND- proteins, with the best model developed with both physicochemical features and a language model/vector representation of protein sequences. The authors verified their model by testing 4 proteins each that were predicted to be COND+ and COND-, with all predictions born out experimentally. The condensate partitioning probability score for COND+ proteins showed a broader distribution of high values than other human proteins, and proteins from various known membraneless organelles (MLOs) also showed broad distributions of high values. Based on these observations, the authors propose that their ML model is a good predictor of condensate partitioning in general, not just partitioning with NPM1. Finally, the authors grouped all human proteins using interaction-based clustering and found that ~3,300 condensate-prone proteins formed 50 clusters, suggesting to the authors that human proteins partition into a larger number of condensate types than currently understood. The authors sought to further verify their findings by showing that protein clusters corresponded to groups of proteins with known associations with each of 7 different MLOs.

The results presented in the ms appear to be of high quality and true and, overall, the authors conclusions are well supported by data. The main finding (prediction) of the work is that a large proportion of human proteins (20%) are likely to partition into biomolecular condensates through protein-protein interactions. The finding that protein interaction clusters formed/weighted using the condensation ML model partitioning probability yields known components of several known MLOs is interesting, and suggests that others of the 50 such clusters may correspond to currently unknown biological condensates. The ML model will be a useful resource for the biomolecular condensates community but needs to made broadly accessible (e.g., through a web resource).

While the computational studies appear to be robust, statistical support for the authors conclusions should be made more clear (see below). While it is interesting that the authors' analyses of condensate partitioning proteins show that they have weaker enrichments of certain physicochemical features compared with proteins that homotypically phase separate, the idea of scaffold and client condensate proteins was previously established by Rosen (see below) and others. There are several points in the manuscript that require clarification and possible revision. With attention to these issues, the manuscript will be a strong candidate for consideration at Nat. Commun.

Major points:

1. To bolster the authors claims regarding the accuracy of their condensation predicting ML model, they should apply it known catalogs of nucleolar proteins to test performance. A nucleolar proteins list was published in Nature by Lamond and Mann in ~2009, and Mann et al. updated this a few years later using higher S/N mass spec methods. Further, Shan et al. recently published high resolution localization of proteins within the different regions of the nucleolus; it may be interesting to test performance for GC proteins, in particular, since NPM1 is a scaffold in this region [Shan L, Xu G, Yao RW, Luan PF, Huang Y, Zhang PH, Pan YH, Zhang L, Gao X, Li Y, Cao SM, Gao SX, Yang ZH, Li S, Yang LZ, Wang Y, Wong CCL, Yu L, Li J, Yang L, Chen LL. Nucleolar URB1 ensures 3' ETS rRNA removal to prevent exosome surveillance. Nature. 2023 Mar;615(7952):526-534. doi: 10.1038/s41586-023-05767-5. Epub 2023 Mar 8. PMID: 36890225.].
2. The authors should perform some level of analysis of the additional partitioning data in the Freibaum/Taylor paper related to stress granules (using G3BP1 as the bait). Are similar insights gained regarding physicochemical feature enrichments for partitioned proteins?
3. The authors verified the performance of their ML model by testing 4 each predicted COND+ and COND- proteins, with performance being 100% accurate, which seems unrealistic. The authors should expand their testing to include a broader range of randomly selected proteins. Maybe they could establish prediction value bins across the full range of values and then randomly select a few proteins from each bin for testing.

Minor points:

1. Page 7. The authors use the term, "universal forces"; what do they mean by these forces? Please rephrase.
2. Fig. 5a. It is difficult to see the data for the different MLOs. Consider using a different format to present these data. Also, please provide some type of quantitation of the distributions (and the statistics of their similarity/difference). Perhaps they could compute a mutual information matrix for all the conditions shown and present the results as a heatmap. Maybe the primary plots could be moved to a supplemental figure.
3. Fig. 5b. What do the interaction networks associated with the 50 clusters on the right look like (e.g., in Cytoscape format)? Are there some proteins with high degree values? These may be scaffold proteins? Do degree values vary with the ML model condensation probability?
4. Fig. 5c. Many details that are needed to aide understanding are missing from the text and legend. What are the units in the circular plots? #s of proteins in a cluster? And what are the different vectors for a given MLO? Different clusters? The criterion for inclusion in a cluster is stated to be two-fold enrichment, but details are not given. Enrichment compared to what? Also, are the clusters dominated by proteins from one MLO, or do some clusters contain proteins from multiple FOs? The authors might consider providing Cytoscape-type graphs of the data shown in Fig. 5c, showing the interactions networks within the different MLOs, but also connections between them when proteins are promiscuous. These could be supplemental data.
5. The authors' observation that proteins that partition with NPM1 have weak feature enrichments relative to proteins that homotypically phase separate echoes Rosen's discussion of scaffold versus client proteins from several years ago, which should be cited [Banani SF, Rice AM, Peeples WB, Lin Y, Jain S, Parker R, Rosen MK. Compositional Control of Phase-Separated Cellular Bodies. Cell. 2016 Jul 28;166(3):651-663. doi: 10.1016/j.cell.2016.06.010. Epub 2016 Jun 30. PMID: 27374333; PMCID: PMC4967043.].

Reviewer #1:

The authors developed a methodology for evaluating the compositional landscape of biomolecular condensates. They divided the condensation-prone proteome into condensate types based on an overlap with previously determined biomolecular interaction profiles to generate a Protein Condensate Atlas. There are several problems:

Reviewer #1: 1. As an atlas resource, the authors need to build a webserver of the resource. So, the users can search, explore, plot, and download the data.

Response: We thank the Reviewer for this comment. Following the Reviewer's suggestion, we have made all the data and the code available on the web via a GitHub repository, enabling the community to explore the data, plot relevant trends etc. To increase the accessibility of this resource further, as a next possible step, we envisage integration with existing biomolecular databases. We will explore this possibility.

Reviewer #1: 2. The authors need to upload the training/test data and the model codes onto GitHub or other publicly available websites.

Response: We thank the Reviewer for this suggestion which we have followed. The data, models and the code associated with the manuscript have now been made available via a GitHub repository that has been referenced in the manuscript. All relevant datasets are included as Supporting Datasets.

Reviewer #1: 3. Are the biomolecular condensates stable? In other words, in which condition, the authors performed the measurement? Will the results be the same when they are measured in other conditions? If they are not stable, there is little meaning to do so. Have the authors tested on various conditions?

Response: We thank the Reviewer for this clarifying question. Condensate systems exist under conditions where demixing is thermodynamically favoured, and, as such, these systems are thermodynamically stable. The location of the demixing binodal can depend on conditions, but once inside the two phase region of the phase diagram, condensates are stable. Following this question, we have expanded on thermodynamic stability of condensate systems in the main text (Introduction - p 1).

Reviewer #1: 4. There are several phase separation prediction models, such as PMID: 37374089. The authors need to make comparisons or discussions of how their model is different and better.

Response: We thank the Reviewer for this question and suggestion. In contrast to the mentioned paper, our approach directly links condensation prediction to protein sequence. This enables estimating the condensation propensity for any sequence rather than only those that have been annotated with GO-terms. Furthermore, by integrating condensation models with biomolecular interaction data, we have constructed an Atlas that predicts the composition of condensate as opposed to assigning one-dimensional propensity scores to proteins. We have highlighted these differences in the revised version of the manuscript (Introduction - p. 2; Results - p. 9).

Reviewer #1: 5. Have the authors evaluated the prediction results with experiments? It seems that the authors did not do any validation during the construction of the Protein Condensate Atlas. They assumed the prediction model can make all the classifications.

Response: This is an excellent point. The predictions have been validated by experiments, using data deposited in the PhaSepDB database. The performance on this validation set has been illustrated in Fig. 6b and 6c (previously Fig. 5e). In the revised version of the manuscript, we have integrated all the validation related analysis to a separate section (p. 9-11) as outlined in more detail in the next comment.

Reviewer #1: 6. With the benchmark dataset, it is easy to build such prediction models. Therefore, the most important contribution was the dataset. The predicted Protein Condensate Atlas should mark the benchmark and predict data. They have different quality levels.

Response: To benchmark our algorithm, we compared the predictions of our Atlas to the data deposited in the PhaSepDB, which describes the composition of known condensate systems. The revised version of the manuscript provides more detailed descriptions of the evaluation approach (Fig. 6a) and expands the validation by investigating the ability of the Atlas to capture sub-MLO level organisation (Fig. 6d and p. 11). Since the composition of many MLOs is unknown, we chose to construct the Protein Condensate Atlas as an unsupervised task to avoid overfitting to available data. We have included this clarification in the text (p. 11). We appreciate the Reviewer's insightful question, which prompted us to emphasise these key design considerations made when constructing the Atlas.

Reviewer #2:

The manuscript describes use of mass spectrometry data for components of an NPM1 (nucleolus-like) condensate to develop a predictor of the components of biomolecular condensates. This is an important goal, given the increasing understanding of the importance of condensates in regulating and organizing biology, and given the experiment challenges in defining components of condensates. The authors utilize machine learning approaches initially to train a nucleolus-specific condensate predictor. They found it was not nucleolus-specific, and the authors supplemented the predictor with protein-protein interaction data from stringDB to add localization/condensate-specificity. Then they applied it to the proteome and referred to the results as a Protein Condensate Atlas. The mass spec data from using NPM1 condensates to generate nucleoli-like condensates by enriching proteins from U2OS cell lysates (Freibaum, B. D., Messing, J., Yang, P., Kim, H. J. & Taylor, J. P. High-fidelity reconstitution of stress granules and nucleoli in mammalian cellular lysate. *J. Cell Biol.* 220, 2021) is a powerful dataset and the authors of this manuscript have used machine learning approaches to extract information about cellular condensates from these mass spec data. They found some interesting general principles regarding the distinction between proteins found to have a high propensity for homotypic phase separation and those proteins found to localize to the NPM1 condensate, some of which are specific to RNA- (or DNA-) containing condensates, such as positive charge and RNA-binding annotation. While the results are unfortunately not clearly interpretable in terms of known components of biomolecular condensates and there is some lack of precision in error analysis and overstatement of results, the approach is intriguing and the conceptual insights and datasets are of value towards addressing the goal of defining the components of biomolecular condensates, an important developing area.

Response: We are grateful for the Reviewer's positive feedback and careful evaluation of our work.

Specific comments:

Reviewer #2: Line 29: The power of proximity labeling approaches is not appropriately recognized here. With large numbers of baits, this approach can be extremely valuable. Of course, as the authors note, it is still candidate driven.

Response: We agree with the Reviewer that proximity labelling is a prominent technique for probing the composition of biomolecular condensates, especially when a large number of baits can be used. We have emphasised its utility in the Introduction (p. 1) of the revised version of the manuscript.

Reviewer #2: Lines 29-35: The problems with characterizing condensates which the authors describe miss the primary concern with any “purification” of condensates changing their composition or dissolving them.

Response: The Reviewer raises an important point. We are working with state-of-the-art data, but they are not free of issues as pointed out. We have now outlined the imperfect nature of the purification processes in the Introduction (p. 2) to give the reader a better insight into the challenges associated with condensate characterisation. The observation that mitochondria-related proteins are to some extent enriched in the condensate fraction (p. 11/12) likely also originates from imperfect purification.

Reviewer #2: The approach driven by NPM1-condensates is biased to nucleic acid-containing condensates. Much of the text needs to be changed to be more specific to NPM1-condensates (as correctly done on lines 152 and 164, 167). For example, line 53 “into condensates” should be “into NPM1-condensates”, line 71 “localisation into condensates” should be “localisation into NPM1-condensates”, line 79 (and 92/93, and 97) “heteromolecular condensates” should be “heteromolecular NPM1-condensates” and line 91 “Condensate composition” should be “NPM1-condensate composition”.

Response: We have incorporated these corrections into the text and appreciate that the Reviewer brought this missing aspect to our attention.

Reviewer #2: Line 65 and Fig 2(c) (and Fig 2(a)). It is assumed that expression level is defined based on the Beck et al mass spec approach referenced in Fig 2(a). This should be clarified in text rather than just with a reference in Fig 2(a).

Response: We have included this clarification and thank the Reviewer for the suggestion.

Reviewer #2: Figure 4. The fact that there is no absolute negative set means that the validation measures are challenging. In addition, components can belong to multiple condensates. This needs to be discussed. The ROC and PRC curves should be shown in supplementary data. In general, there does not appear to be a serious evaluation of the error rate of their predictor (and therefore the atlas).

Response: The Reviewer is correctly pointing out the challenges around choosing relevant validation metrics in the absence of an absolute negative set. This is why we did not use ROC and PRC curves as they would require a test set with labels. Instead, we quantified performance by calculating the enrichment factor for each predicted condensate cluster. This factor describes the degree to which the cluster is enriched in proteins that are experimentally found to localise in the same condensate compared to random sampling. We have now added a figure panel to our manuscript to highlight how this validation process is performed (Fig. 6a).

Reviewer #2: The cross-validation procedure with the ROC analysis appears to be biased. The authors wrote (lines 247-250): "The optimal set of hyperparameters for each model was determined in a 10-fold cross-validation process by requiring the area of the receiver operator characteristic curve (auROC) to be at its maximum. Within each fold, 80:20 split between the train and the validation data was used." If the authors used the validation split to optimize hyperparameters, the auROCs estimated on those validation splits are going to be over-estimated. To report unbiased auROCs, they need to split off unseen "test" data that is not used in either parameter or hyperparameter optimization.

Response: We have now conducted the analysis in the manner suggested by the Reviewer where a separate set is put aside for testing and have shown the predictive power on the unseen test set in Figure 3b-c. We thank the Reviewer for suggesting this change.

Reviewer #2: The results in Figure 5(a) are not strong with most of the proteins annotated in the various condensates having a prediction around 0.2, near the peak of the curve for proteins with no annotation. The authors do not acknowledge this. Given this, what is the meaning of the predictor score?

Response: We thank the Reviewer for this question and comment. We confirm that the scores for the proteins in the highlighted MLOs differ significantly from the scores of proteins with no condensate annotation. We agree that the previous plots poorly conveyed this difference. In the revised manuscript, we compared the percentiles at which the predictions lie (rather than absolute scores) and replaced the previous panel with one that shows the distributions independently (Fig. 5a). This updated panel should better highlight the difference. Additionally, we have compared the key statistics of the distributions (median, 75th percentile, and 90th percentile) to confirm their differences (Supplementary Fig. S2).

Reviewer #2: Lines 181/182: "This process resulted in the generation of over 230 clusters. Around 50 of these had at least half of the proteins in the cluster condensation-prone, suggesting they may constitute condensates." The authors should consider the possibility that other clusters besides the 50 listed in supplementary dataset 7 could represent condensates and should also list the components within all 230 clusters in a supplementary dataset.

Response: We have now listed the compositions of all these clusters in the SI (Supplementary Dataset 9).

Reviewer #2: Fig 5(c) is very unclear. The message (lines 185/186: "Remarkably, we found two thirds of the clusters to show over two-fold enrichment in proteins that have been experimentally seen to co-localise into the same condensate system") is not supported since it is not obvious how the comparison

is being made. What are these plots and how are they generated? Also, why is this enrichment "remarkable" considering that the protein-protein information from stringDB was included in making these clusters. Since proteins are more likely to interact with other co-localized proteins, wouldn't this observation be expected?

Response: We found the strong enrichment intriguing, considering that the interaction data in StringDB does not describe what happens above the saturation concentrations where biomolecules localise into condensates. We have removed the word "remarkable" from the text and instead replaced it with this reasoning (p. 11). Additionally, we have highlighted in the text that the dataset used for validation is an independent test set that was not used as input data for constructing the Atlas (p. 9).

Reviewer #2: The claim to have generated a protein condensate atlas is not really supported. There is no clear list of clusters that are supposed to actually match specific described condensates. The list in Supplementary dataset 7 is not annotated with suggested biomolecular condensates "matching" each or potentially included within the clusters given.

Response: To address this limitation, we have now highlighted in Supplementary Dataset 9 (previously #7) which cluster overlaps with known condensate systems and which ones do not overlap with any characterised system. This makes this information easily retrievable. We appreciate the Reviewer's comment, which we feel has helped us further improve the quality of our work.

Reviewer #2: The authors need to acknowledge that specific protein components are often part of more than one biomolecular condensate. Some obvious examples are proteins that have some degree of nuclear localization and some cytoplasmic localization, and are found in distinct condensates in these environments. Other examples include proteins that are found in highly related RNA "granules" including P bodies, stress granules and mRNA transport granules.

Response: This is an excellent point. We have now highlighted in the text that each protein belongs to only a single cluster (p. 8). The Reviewer is correct about related condensates showing overlaps in their composition. We have now specifically shown that this observation is in agreement with our results presented in Figure 6c and Supplementary Table 8 where some clusters show strong enrichment for multiple and often functionally related MLOs. We have highlighted this argument in the text (p. 11). We are grateful to the Reviewer for this comment that prompted us to highlight these results.

Minor points:

Reviewer #2: Multiple figures used “density” as a y-axis label without clear definition of what this means. Density is likely dependent on the bin size for the x-axis values but it is hard to match these numbers to the total values being plotted.

Response: We have specified that in these cases the y-axis labels refer to the kernel density estimate.

Typographical errors:

Line 119 “suggested the bind RNA” should be “suggested they bind RNA”

Line 158 “they may part of condensates” should be “they may be part of condensates”

Line 168 “is through to be driven by universal forces” should be “is thought to be driven by universal forces”

Line 166 “Althought the model” should be “Although the model”

Line 174 “scores for these proteins that did not have any” should be “scores for those proteins that did not have any”

Fig 3(a) legend: “phase separation propnesity” should be “phase separation propensity”

Response: We have introduced these corrections and thank the reviewer for highlighting them.

Reviewer #3 (Remarks to the Author):

The article by Saar, et al., addresses relationships between protein sequences and associated physicochemical features, and protein localization within cellular biomolecular condensates. The authors utilized a previously published protein condensates list (COND+ proteins) generated using the nucleolar protein, NPM1, and U2OS cell lysate. The authors augmented this list with a COND- list generated using other existing data on all proteins in U2OS cells. The authors showed that protein partitioning into the condensate fraction was independent of protein concentration in lysate, and that some physicochemical properties of proteins were enriched or depleted in the COND+ group. They also noted that COND+ proteins were less enriched in phase separation-associated features than proteins known to undergo homotypic phase separation, with their interpretation being that condensate partitioning and autonomous phase separation are driven by different sets of protein features. The authors next showed that COND+ proteins often bind RNA, and that the physical properties of RNA-binding and non-RNA-binding COND+ proteins are somewhat different. Using StringDB, the authors showed that COND+ proteins experience larger numbers of interactions than COND- proteins, and are most likely to interact with other COND+ proteins. At this point in the ms, the authors propose that proteins partition into condensates either through protein-RNA or protein-protein interactions. The authors next developed machine learning (ML) models to predict condensation behavior using various combinations of feature- and sequence-based properties of COND+ and COND- proteins, with the best model developed with both physicochemical features and a language model/vector representation of protein sequences. The authors verified their model by testing 4 proteins each that were predicted to be COND+ and COND-, with all predictions born out experimentally. The condensate partitioning probability score for COND+ proteins showed a broader distribution of high values than other human proteins, and proteins from various known membraneless organelles (MLOs) also showed broad distributions of high values. Based on these observations, the authors propose that their ML model is a good predictor of condensate partitioning in general, not just partitioning with NPM1. Finally, the authors grouped all human proteins using interaction-based clustering and found that ~3,300 condensate-prone proteins formed 50 clusters, suggesting to the authors that human proteins partition into a larger number of condensate types than currently understood. The authors sought to further verify their findings by showing that protein clusters corresponded to groups of proteins with known associations with each of 7 different MLOs.

The results presented in the ms appear to be of high quality and true and, overall, the authors conclusions are well supported by data. The main finding (prediction) of the work is that a large proportion of human proteins (20%) are likely to partition into biomolecular condensates through protein-protein interactions. The finding that protein interaction clusters formed/weighted using the condensation ML model partitioning probability yields known components of several known MLOs is interesting, and suggests that others of the 50 such clusters may correspond to currently unknown biological condensates. The ML model will be a useful resource for the biomolecular condensates community but needs to be made broadly accessible (e.g., through a web resource). While the computational studies appear to be robust, statistical support for the authors conclusions should be made more clear (see below). While it is interesting that

the authors' analyses of condensate partitioning proteins show that they have weaker enrichments of certain physicochemical features compared with proteins that homotypically phase separate, the idea of scaffold and client condensate proteins was previously established by Rosen (see below) and others. There are several points in the manuscript that require clarification and possible revision. With attention to these issues, the manuscript will be a strong candidate for consideration at Nat. Commun.

Response: We appreciate the Reviewer's positive feedback and their thorough review of our manuscript.

Major points:

Reviewer #3: 1. To bolster the authors claims regarding the accuracy of their condensation predicting ML model, they should apply it known catalogs of nucleolar proteins to test performance. A nucleolar proteins list was published in Nature by Lamond and Mann in ~2009, and Mann et al. updated this a few years later using higher S/N mass spec methods. Further, Shan et al. recently published high resolution localization of proteins within the different regions of the nucleolus; it may be interesting to test performance for GC proteins, in particular, since NPM1 is a scaffold in this region [Shan L, Xu G, Yao RW, Luan PF, Huang Y, Zhang PH, Pan YH, Zhang L, Gao X, Li Y, Cao SM, Gao SX, Yang ZH, Li S, Yang LZ, Wang Y, Wong CCL, Yu L, Li J, Yang L, Chen LL. Nucleolar URB1 ensures 3' ETS rRNA removal to prevent exosome surveillance. Nature. 2023 Mar;615(7952):526-534. doi: 10.1038/s41586-023-05767-5. Epub 2023 Mar 8. PMID: 36890225.].

Response: Thank you for this excellent suggestion. We have now compared the predictions of our Atlas to these published data and found the described nucleolar regions to be enriched into distinct condensate clusters within our Atlas. We have included this analysis and a summary of the findings in the revised version of the manuscript (p. 10-11, Fig 6c, Supplementary Fig. 6).

Reviewer #3: 2. The authors should perform some level of analysis of the additional partitioning data in the Freibaum/Taylor paper related to stress granules (using G3BP1 as the bait). Are similar insights gained regarding physicochemical feature enrichments for partitioned proteins?

Response: We have incorporated an analysis of the reconstituted G3BP1 condensates in the revised version of the manuscript (p. 4). The results show similar trends in physicochemical feature enrichment (Supplementary Fig. 1). We thank the Reviewer for suggesting this additional analysis.

Reviewer #3: 3. The authors verified the performance of their ML model by testing 4 each predicted COND+ and COND- proteins, with performance being 100% accurate, which seems unrealistic. The authors should expand their testing to include a broader range of randomly selected proteins. Maybe they could

establish prediction value bins across the full range of values and then randomly select a few proteins from each bin for testing.

Response: This is a great suggestion. We have expanded the comparison to a larger protein set. Results from this expanded set are highlighted in Supplementary Dataset 7 and are summarised in the manuscript (p. 6).

Minor points:

Reviewer #3: 1. Page 7. The authors use the term, “universal forces”; what do they mean by these forces? Please rephrase.

Response: We have revised this sentence to improve its clarity and appreciate the Reviewer’s feedback.

Reviewer #3: 2. Fig. 5a. It is difficult to see the data for the different MLOs. Consider using a different format to present these data. Also, please provide some type of quantitation of the distributions (and the statistics of their similarity/difference). Perhaps they could compute a mutual information matrix for all the conditions shown and present the results as a heatmap. Maybe the primary plots could be moved to a supplemental figure.

Response: Prompted by the Reviewer’s comment, we have presented these data as individual distributions to better highlight the differences (Fig. 5a). We also compared the key characteristics of the distributions (median, 75th and 90th percentile) to identify significant differences (Supplementary Fig S2). We thank the Reviewer for their suggestion to review how the data is visualised. We do not think that a mutual information matrix calculation makes sense. Our goal is to highlight that the model predicts higher scores for proteins in different MLO types than for proteins with no MLO annotation. This objective can be achieved by directly comparing the distributions as we have done.

Reviewer #3: 3. Fig. 5b. What do the interaction networks associated with the 50 clusters on the right look like (e.g., in Cytoscape format)? Are there some proteins with high degree values? These may be scaffold proteins? Do degree values vary with the ML model condensation probability?

Response: Following the Reviewer’s suggestion, we compared the degree values between proteins in different categories and indeed observed differences in degree values. We have included this analysis in the revised version of the manuscript (Fig. 5c-d). We have also linked the finding to a similar trend that we had seen for the NPM1-reconstituted condensates in the earlier part of the manuscript (p. 9 and Fig.

3c). Additionally, as suggested, we have highlighted the interactions in a Cytoscape-type graph for an exemplary condensate cluster in Supplementary Fig. S3.

Reviewer #3: 4. Fig. 5c. Many details that are needed to aid understanding are missing from the text and legend. What are the units in the circular plots? #s of proteins in a cluster? And what are the different vectors for a given MLO? Different clusters? The criterion for inclusion in a cluster is stated to be two-fold enrichment, but details are not given. Enrichment compared to what? Also, are the clusters dominated by proteins from one MLO, or do some clusters contain proteins from multiple FOs? The authors might consider providing Cytoscape-type graphs of the data shown in Fig. 5c, showing the interactions networks within the different MLOs, but also connections between them when proteins are promiscuous. These could be supplemental data.

Response: We have added substantial detail regarding how the enrichment values are calculated (p. 9), including a separate panel that focuses on this aspect (Fig. 6a). We have replaced the star-diagram based visualisations with 2D matrix (Fig. 6c) that conveys the same information and hopefully in a clearer format. The Reviewer is correct about some clustered resembling multiple MLOs. We have explained this aspect in the revised manuscript (p. 11). Such clusters have now been annotated in Supporting Dataset 8. Following the Reviewer's suggestion, we have included the Cytoscape-type graphs (Supplementary Fig. S4) to visualise the enrichments. We are very grateful to the Reviewer for raising these insightful questions, which have led to these improvements, significantly enhancing the quality of our manuscript.

Reviewer #3: 5. The authors' observation that proteins that partition with NPM1 have weak feature enrichments relative to proteins that homotypically phase separate echoes Rosen's discussion of scaffold versus client proteins from several years ago, which should be cited [Banani SF, Rice AM, Peeples WB, Lin Y, Jain S, Parker R, Rosen MK. Compositional Control of Phase-Separated Cellular Bodies. *Cell*. 2016 Jul 28;166(3):651-663. doi: 10.1016/j.cell.2016.06.010. Epub 2016 Jun 30. PMID: 27374333; PMCID: PMC4967043.].

Response: We agree with this suggestion and have referenced it in the manuscript when discussing the results (p. 4).

Reviewers' Comments:

Reviewer #2:

Remarks to the Author:

The revised version of the manuscript reasonably addresses the previous concerns and comments. This version is much more clear, including the figures and the validation methods. It also provides tables with compositional predictions for known condensates as well as suggestions for compositions of other condensates. The use of PhaSepDB as the definition of the "currently" known condensates is somewhat limiting, and should be acknowledged. In general, though, the work is a significant contribution to the field of biomolecular condensates, based on moving beyond simple predictors of phase separation of individual proteins to prediction of composition of a sizeable set of biomolecular condensates.

Reviewer #3:

Remarks to the Author:

The revised article by Saar, et al., addresses this reviewer's prior comments and seems to also address those of the other reviewers (with the exception of establishing a web-based portal for the Atlas). The article reports a novel approach to clustering groups of proteins predicted to localize within condensates and will be a valuable resource for the biomolecular condensates community. The article is suitable for publication in Nat. Commun.

The authors are encouraged to address the minor point below to improve the clarity of their manuscript.

1. Fig. 2d. This statement in the legend for panel d is confusing: "The proteins that partitioned into condensates (COND+; green) had higher hydrophobicity per molecular weight than those that did not (COND-; red)." The values for the COND+ protein set are more negative than those for the COND- set, meaning that the COND+ proteins are less hydrophobic than the COND- proteins; correct?

This comment also pertains to lines 77-78 of the text: "We noticed that proteins with a high tendency to localise into the condensates (COND+; green) were more hydrophobic ($p < 0.0001$, Mann-Whitney U test; Figure 2d) ..."

In general, proteins that form condensates are not enriched in hydrophobic residues (e.g., as reported for fusion oncoproteins in Tripathi, et al., Nat. Commun., 2023).

2. SFig. 3b. Please include an x-axis label.

3. Line 109, type-o. "... divided both the COND+ and COND- datasets into two groups..." should be "... divided both the COND+ and COND- datasets into two groups..."

4. Lines 109-112. "By calculating p-values (Mann-Whitney U-test) for several key biophysical features between the two groups (Figure 3b), we observed that sequence length and hydrophobicity were the primary factors influencing partitioning in non-RNA-binding proteins (non-RBP; blue). In contrast, these factors played a less prominent role for RNA-binding proteins (RBP; red)." Can the authors please clarify whether the COND+ non-RNA binding proteins are more or less hydrophobic than the COND- set. Reporting just the p-value doesn't address the magnitudes of the respective hydrophobicity values.

5. Lines 169-170. Type-o. "Taken together, these result clearly demonstrates the ability of the model to extend beyond the sequence space covered in the training set." Should be "Taken together, these results clearly demonstrate the ability of the model to extend beyond the sequence space covered in the training set."

6. Lines 248-250. "Using a similar protocol as described above, we calculated the enrichment values for all the clusters with respect to the characterised subregions within the nucleus and

found the maximum enrichment values to be 50 or above for most of the regions (Supplementary Figure 5).” The middle phrase should probably read, “... characterised subregions within the nucleolus and found...”, not “nucleus”.

7. Fig. 6 c and d. Are the same proteins represented in each MLO column? This detail should be added to the figure legend. The authors should label a few hallmark MLO proteins to aid comprehension of the figures. How was the list of displayed proteins established?

REVIEWER COMMENTS

Reviewer #2

Reviewer #2 (Remarks to the Author): The revised version of the manuscript reasonably addresses the previous concerns and comments. This version is much more clear, including the figures and the validation methods. It also provides tables with compositional predictions for known condensates as well as suggestions for compositions of other condensates. The use of PhaSepDB as the definition of the "currently" known condensates is somewhat limiting, and should be acknowledged. In general, though, the work is a significant contribution to the field of biomolecular condensates, based on moving beyond simple predictors of phase separation of individual proteins to prediction of composition of a sizeable set of biomolecular condensates.

Response: We thank the Reviewer for a thorough review of the revised version of our manuscript and for the supportive comments. We have mentioned that PhaSepDB is limited in its coverage of condensate types (p. 9). We thank the Reviewer for suggesting the inclusion of this clarification.

Reviewer #2 (Remarks on code availability): It appears that the github is not public as I was unable to see the code. Perhaps the authors don't want to make it public yet but then they should provide another website accessible to the reviewers.

Response: We have made the github repository publicly available.

Reviewer #3

Reviewer #3 (Remarks to the Author): The revised article by Saar, et al., addresses this reviewer's prior comments and seems to also address those of the other reviewers (with the exception of establishing a web-based portal for the Atlas). The article reports a novel approach to clustering groups of proteins predicted to localize within condensates and will be a valuable resource for the biomolecular condensates community. The article is suitable for publication in Nat. Commun.

Response: We thank the Reviewer for reviewing the revised version of our manuscript and for their supportive comments.

The authors are encouraged to address the minor point below to improve the clarity of their manuscript.

Reviewer #3: 1. Fig. 2d. This statement in the legend for panel d is confusing: “The proteins that partitioned into condensates (COND+; green) had higher hydrophobicity per molecular weight than those that did not (COND-; red).” The values for the COND+ protein set are more negative than those for the COND- set, meaning that the COND+ proteins are less hydrophobic than the COND- proteins; correct? This comment also pertains to lines 77-78 of the text: “We noticed that proteins with a high tendency to localise into the condensates (COND+; green) were more hydrophobic ($p < 0.0001$, Mann–Whitney U test; Figure 2d) ...” In general, proteins that form condensates are not enriched in hydrophobic residues (e.g., as reported for fusion oncoproteins in Tripathi, et al., Nat. Commun., 2023).

Response: The Reviewer is correct, proteins in the COND+ dataset are less hydrophobic and not enriched in hydrophobic residues. We have corrected these two sentences and thank the Reviewer for raising this error to our attention.

Reviewer #3: 2. SFig. 3b. Please include an x-axis label.

Response: We have added the axis label and thank the Reviewer for highlighting its absence.

Reviewer #3: 3. Line 109, type-o. “... divided both the COND+ an COND- datasets into two groups...” should be “... divided both the COND+ and COND- datasets into two groups...”

Response: We have corrected this typographical error and thank the Reviewer for raising it to our attention.

Reviewer #3: 4. Lines 109-112. “By calculating p-values (Mann-Whitney U-test) for several key biophysical features between the two groups (Figure 3b), we observed that sequence length and hydrophobicity were the primary factors influencing partitioning in non-RNA-binding proteins (non-RBP; blue). In contrast, these factors played a less prominent role for RNA-binding proteins (RBP; red).” Can the authors please clarify whether the COND+ non-RNA binding proteins are more or less hydrophobic than the COND- set. Reporting just the p-value doesn't address the magnitudes of the respective hydrophobicity values.

Response: The Reviewer raises a great point. We have updated the figure to show both the effect size with a direction and the significance. Non-RNA binding condensate partitioning proteins have lower hydrophobicity values than non-RNA binding proteins that do not partition into condensates. We have added this comment to the manuscript (p. 4).

Reviewer #3: 5. Lines 169-170. Type-o. “Taken together, these result clearly demonstrates the ability of the model to extend beyond the sequence space covered in the training set.” Should be “Taken together, these results clearly demonstrate the ability of the model to extend beyond the sequence space covered in the training set.”

Response: We have corrected this typographical error and thank the Reviewer for highlighting it.

Reviewer #3: 6. Lines 248-250. “Using a similar protocol as described above, we calculated the enrichment values for all the clusters with respect to the characterised subregions within the nucleus and found the maximum enrichment values to be 50 or above for most of the regions (Supplementary Figure 5).” The middle phrase should probably read, “... characterised subregions within the nucleolus and found...”, not “nucleus”.

Response: The Reviewer is correct. We have replaced nucleus with nucleolus in this sentence. We thank the Reviewer for bringing the error to our attention.

Reviewer #3: 7. Fig. 6 c and d. Are the same proteins represented in each MLO column? This detail should be added to the figure legend. The authors should label a few hallmark MLO proteins to aid comprehension of the figures. How was the list of displayed proteins established?

Response: This is a great suggestion. We have highlighted some of the hallmark proteins to aid the comprehension of the figures (Fig. 6c and p. 10). The proteins displayed in each column are the same, they represent 8300 proteins across the 62 predicted condensate clusters in the Atlas (Fig. 6b).